# Hop (*Humulus lupulus* L.) Phenolic Compounds Profile Depends on Cultivar and Plant Organ Maturity

**DOI:** 10.3390/molecules30112365

**Published:** 2025-05-29

**Authors:** Jakub Piekara, Dorota Piasecka-Kwiatkowska, Hanna Hołaj, Małgorzata Jędryczka, Oluwafemi Daniel Daramola, Krzysztof Dwiecki

**Affiliations:** 1Department of Food Biochemistry and Analysis, Poznan University of Life Sciences, Mazowiecka 48, 60-623 Poznań, Poland; dorota.piasecka-kwiatkowska@up.poznan.pl (D.P.-K.); 89295@student.up.poznan.pl (O.D.D.); 2Agricultural Experimental Station ‘Jastków’, Panieńszczyzna, Chmielowa 5, 21-002 Jastków, Poland; rzd.jastkow@iung.pulawy.pl; 3Institute of Plant Genetics, Polish Academy of Sciences, Strzeszyńska 34, 60-479 Poznań, Poland; mjed@igr.poznan.pl

**Keywords:** polyphenols, flavonoids, phenolic acids, hop, plant organ, cones, leaves, stalks, zero waste policy

## Abstract

Hop by-products constitute a significant part of biomass in cones production for the brewing industry. The phenolic compounds (PCs) they contain can be used in the food and pharmaceutical industries but require qualitative and quantitative analysis. The aim of this study was to investigate the extent to which phenolic compounds profiles depend on cultivar, plant organ, and plant level. This paper shows for the first time that for hop, it is not only the plant organ that is important for PC content, but also the level from which it is obtained. Metabolites were investigated in cones, leaves, and stalks at three levels of the plant in Polish hop cultivars (Marynka, Lubelski, and Magnum). The PC content showed a differentiation due to the cultivar of hops, their anatomical part, and position in the plant (level), which reflects the degree of organ maturity. The total PC was the highest in leaves (up to 922 mg/100 g), while lower contents were found in cones (up to 421 mg/100 g) and stalks (up to 105 mg/100 g). The main PCs of leaves were kaempferol-3-glucoside (up to 328 mg/100 g) and rutin (up to 293 mg/100 g), while rutin dominated in cones (up to 209 mg/100 g).

## 1. Introduction

Hop cones, traditionally associated with the brewing industry due to their valuable flavor and aroma constituents, represent one of the richest natural sources of phenolic compounds (PCs). Their efficacy in the prevention and treatment of various diseases is well documented in the scientific literature. These properties included sedative effects, blood pressure reduction, and antioxidant and estrogenic activities. Hop plant extracts are commonly used in the form of dietary supplements and as ingredients in cosmetic products [1,2,3,4].

Until now, the most valued compounds of hop were α-bitter acids (such as cohumulone and humulone), β-bitter acids (such as colupulone and lupulone), and essential oils (like α-humulene) from cones, which were largely used in the production of beverages. However, some interest was also focused on leaves and stalks, which were considered until recently to be waste. The newly introduced “zero waste” policy makes plant residues an interesting material for food products and additives of high nutritional value [5]. Only 15% of the processed hop plant is utilized in brewing for beer production, while the rest is returned to the fields as partially fermented biomass. An essential component of the production chain is enhancing the value of hops by providing scientific evidence of the potential of brewing waste as a source of bioactive compounds for further processing [6,7].

The hop plant contains numerous valuable flavonoids and phenolic acids (PAs) that contribute to its non-enzymatic antioxidant system. These compounds hold potential for application in the food industry, functional foods, pharmaceuticals, and dietary supplements. They have been effectively utilized in the treatment of conditions such as depression, insomnia, joint inflammation, and allergies, as well as in supporting hormone replacing therapy (HRT) in postmenopausal women [8,9,10,11]. In addition to supporting and preventing civilization diseases, hop compounds, especially xanthohumol (XN), are used in the treatment of cancerous lesions, since traditional chemotherapy leaves many side effects and is often insufficient for some forms of cancer, leading to poor quality of patient life. The main action of PCs in cancer is to prevent inflammation by inhibiting the proliferation of cancer cells, without side effects [12]. Another important function in disease prevention is demonstrated by quercetin, which can build into the intestinal walls, improving their integrity and preventing “leaky gut”. This in turn affects the number of enterocytes necessary for proper absorption of nutrients by creating a favourable environment for intestinal microflora [13]. In addition, PCs are used in construction, agriculture, and environmental engineering as impregnates, as well as as antifungal and antiparasitic agents [5].

Flavonoids refer to a group of compounds made by two benzoic rings connected by a heterocyclic pyrone ring. Many of them have so far been identified in hop cones and products obtained from them (like hops pellets). These include: antocyanins, prenylated chalcones (XN), dihydrochalcones, dihydroflavonols, flavanols (procyanidin dimers, epicatechins, and epigallocatechins), flavanones (naringenin and hesperetin derivatives), flavones (apigenin and chrysoeriol glycosides), flavonols (quercetin, kaempferol, and myricetin glycosides) and isoflavonoids [14,15,16]. Among PAs, the presence of gentisic acid (GNT), ferulic acid (FA), caffeic acid (CA), chlorogenic acid (CGA), vanillic acid (VA), gallic acid (GAL), o-coumaric acid (oCA), p-coumaric acid (4CA), cinnamic acid, syringic acid, p-hydroxybenzoic, and ellagic acid in hop cones has been recorded [14,17,18,19]. PCs in hops most often occur in the form of glycosides or other conjugates, such as kaempferol 3-O-xylosyl-glucoside (K-3-G) or rutin (RUT) [14,16]. Not all phenolic compounds found in hop cones are present in high concentrations, and their levels are influenced by various factors, including cultivar, growing conditions, viral infections, and the age of the plant. For example, according to [16], the content of RUT in the hop cones of the Lubelski cultivar is almost 2.5 times higher than in the Magnum variety.

The biological function of phenolics depends on the quantity of OH groups, double bonds, and polymerization grade. The concentration of PCs depends on many factors such as the soil, sunlight, parasites, insects, and herbivores. The capacity of plants that contain PCs results from the response to abiotic and biotic conditions, which is genetically determined. PCs perform the role of free radical scavenger, such as secondary antioxidants. Their concentration arises with adverse environmental conditions. The primary factor that induces an increase in the concentration of phenolic compounds (PCs) is oxidative stress, which in plants is largely triggered by exposure to ultraviolet (UV) radiation. PCs, due to the presence of OH groups, have the capacity to store an excess of absorbed energy in the chloroplasts that reduce the amount of reactive oxygen [20,21,22].

The aim of this work was to determine the profile of PCs in different plant organs, such as cones, leaves, and stalks at three levels of the plant. The purpose in question arises from the height of the hop plants, which are extremely high—up to 10 m—so we hypothesized that the contents of various PCs may greatly vary between samples taken from different levels of the plant due to different cells’ metabolism levels. The plan is to divide the plant into sections in terms of both the height and anatomical plants part of three-hop cultivars grown in Poland and bred for Polish environmental conditions. The study was done using plants from ecological-type cultivation, with minimized use of chemical interventions (fungicides, insecticides, etc.) and growing on eco-supports without the use of toxic cresols. To our knowledge, the content of PCs has been studied in cones only and has not been studied in the leaves and stalks of hops. Moreover, the PC contents in cones were studied for the average cone biomass, without the separation to different plant levels. Our proposed expanded research on PC contents in hops can be very useful for understanding the location of bioactive compounds in hops for the better utilization of plant by-products.

## 2. Results and Discussion

### 2.1. Differences Between Cultivars

Anatomical parts that play different roles in plants are characterized by distinct structures and, therefore, chemical compositions. In the case of PCs, the strength of their binding to tissues is also varied. For this reason, alternative methods of PCs extraction may be necessary for various anatomical parts. In this study, extractions with 70% methanol with acid hydrolysis and 80% methanol without hydrolysis were used. It was found that the method with hydrolysis was optimal for the extraction of FA and GNT from stalks. On the other hand, other PCs were more efficiently extracted with 80% methanol without hydrolysis. This is related to the phenomenon of binding PAs to the cell walls of stalks. FA and its oligomers are important components of the plant cell walls, forming cross-links between polysaccharides, polysaccharides and lignin, and polysaccharides and proteins [23]. Also, hydroxybenzoic acids, which include GNT, are often present in bound form, being component of lignins and hydrolysable tannins [24].

PAs (GAL, 4CA, FA, CA, protocatechuic acid, GNT, CGA, and sinapic acid (SA)) as well as flavonoids (RUT and K-3-G) have been identified in various anatomical parts of hops (Table 1, Table 2, Table 3, Table 4, Table 5, Table 6, Table 7, Table 8 and Table 9). Additionally, two PCs were identified as flavonoids based on their characteristic UV-Vis (ultraviolet–visible) absorption spectra (absorption maxima at 255 nm and 354 nm for flavonoid 1—Appendix A—as well as 264 nm and 345 nm for flavonoid 2—Appendix A) (Appendix A). On the basis of comparison with the literature data, we found that the absorption bands with a maximum at wavelengths 255 nm and 354 nm (flavonoid 1) are characteristic for flavones (like luteolin or diosmetin) and the bands with a maximum at wavelengths 264 nm and 345 nm (flavonoid 2) are typical for flavonols. The first absorption maximum was attributed to the ring A of the flavonoid structure, while the second corresponded to the ring B and the connecting system in flavonoids [25]. However, their exact structure has not been recognized.

In order to compare the content of total PCs in different hop cultivars, the most mature lower parts of the plants were analyzed. The basic raw materials obtained from the hops were the cones. The content of total PCs in the lower cones was found to be 357.59 mg/100 g (Marynka), 271.90 mg/100 g (Lubelski), and 61.68 mg/100 g (Magnum, Table 1) respectively. Similarly, in the lower leaves, the highest content was found in Marynka (246.94 mg/100 g), which was followed by Lubelski (167.63 mg/100 g) and Magnum (124.68 mg/100 g, Table 4). When the stalks were analyzed, the highest content of total PCs was noticed in Lubelski 78.98 mg/100 g, whereas this value in Magnum and Marynka was similar, averaging about 65 mg/100 g (Table 7). The analysis of flavonoid content in the lower hops cones confirmed that the highest content occurred in Marynka (274.31 mg/100 g; see Table 1). A slightly lower level was present in Lubelski (211.70 mg/100 g), while Magnum was poor in flavonoids (31.00 mg/100 g), and PAs accounted for almost half of PCs in this cultivar (30.68 mg/100 g; see Table 1). The content of PAs in the lower cones of the three cultivars was the highest in Marynka (83.27 mg/100 g; see Table 1), which was followed by Lubelski and Magnum (60.20 mg/100 g and 30.68 mg/100 g, respectively; see Table 1). The predominant lower (mature) cones’ phenolic compound in all cultivars was RUT, where the highest content was observed in Marynka (184.18 mg/100 g), which was followed by Lubelski and Magnum (78.62 mg/100 g and 15.68 mg/100 g, respectively; see Table 1). The level of flavonoids in the lower leaves of the analyzed cultivars reflects the total content of PCs (178.43 mg/100 g in Marynka, 120.27 mg/100 g in Lubelski, and 89.93 mg/100 g in Magnum; see Table 4). In the same order, the PAs were ranked (68.51 mg/100 g, 47.36 mg/100 g, and 34.75 mg/100 g in Marynka, Lubelski, and Magnum respectively; see Table 4). The main PCs found in the lower leaves of all cultivars were K-3-G and RUT. The level of these compounds was equal to 92.21 mg/100 g and 62.65 mg/100 g for Marynka, 75.66 mg/100 g and 19.80 mg/100 g for Lubelski, and 50.04 mg/100 g and 27.07 mg/100 g for Magnum, respectively, (Table 4). In the lower hops stalks, two flavonoids were found using the 80% methanol extraction. The predominant was RUT, with the highest value recorded in Lubelski (26.10 mg/100 g), which was followed by Magnum and Marynka (17.77 mg/100 g and 16.12 mg/100 g, respectively; see Table 7). In turn, the level of K-3-G was in the range from 6.75 mg/100 g to 8.68 mg/100 g, and no statistically significant differences between cultivars were found. In this anatomical part, the prevailing PAs were GNT in Marynka (12.07 mg/100 g), CA in Lubelski (14.57 mg/100 g), and GAL in Magnum (16.28 mg/100 g; see Table 7). In turn, in the cones and leaves, CA clearly predominated among the PAs.

A significantly higher total phenolic compound content was determined using the Folin–Ciocalteu (FC) method, with the highest levels observed in the cones of the Marynka and Magnum cultivars (2690 mg/100 g and 2540 mg/100 g, respectively); see Figure 1. Kowalczyk et al. [26] reported a greater total phenolic contents, with values of 3874 mg/100 g for the Magnum cultivar and 5325 mg/100 g for Marynka, as determined in cone extracts obtained using 50% methanol. These authors also quantified total flavonoid levels equal to 268 mg/100 g (Magnum) and 425 mg/100 g (Marynka). Differences between the results obtained by Kowalczyk et al. [26] and our group may be attributed to several factors. They could be the effects of different growing conditions (insolation, precipitation, soil, fertilization, etc.), viral infections, age of plants, and in some cases the cultivars themselves. The total PCs determination using the FC method and flavonoids using the aluminium ion complexation method can yield surprising results. In particular, this is true in the case of the FC method, because not only PCs but also several interference substances with reducing properties may react with this reagent or/and with PCs. Additionally, the results of these determinations were presented as one standard compound—gallic acid equivalent (GAE)—in the FC method and quercetin in the Al ion method. In the case of our studies, individual standards listed in Table 1, Table 2, Table 3, Table 4, Table 5, Table 6, Table 7, Table 8 and Table 9 were used for the determination of flavonoids and PAs. The observed differences may be also due to the extraction technique used.

Quercetin, kaempferol, and their glycosides (RUT and kaempferol-3-hexoside) were identified in hops by Bilska et al. and Magalhães et al. [16,17]; according to these authors, kaempferol and quercetin are not present in hops in free form but are glycosidically bound with sugar moieties in position 3. In our analysis, these compounds include K-3-G and RUT (quercetin-3-rutinoside). Jelinek et al. [18] claimed that hop cones (Sládek, Saaz, and Taurus cultivars) contain RUT, catechin, epicatechin, coumarin, and GNT. According to these authors, the PC contents in cones depend on the cultivar and age of the plant. For the Saaz cultivar, the total PC contents ranged from 690 to 830 mg/100 g (8-year-old plants) and 570 to 590 mg/100 g (13-year-old plants). In the case of the Taurus cultivar, this value was equal to 170 mg/100 g (1-year-old plants) and 140 mg/100 g (8-year-old plants). Viral infections were also found to increase the contents of PCs. The dominant PCs in the Saaz cultivar were catechin (43.88–49.14% of all PCs) and GNT (19.63–21.42% of all PCs). In the case of the Taurus cultivar, RUT was the prevailing PC (24.99–34.20% of phenolics), which was followed by GNT (14.09–16.36%). The dominant share of RUT was confirmed in our study in the cultivars Marynka and Lubelski. In turn, GNT was not present in the cones but constituted the main phenolic acid (PA) compound in the stalks of the Marynka and Magnum hops.

### 2.2. Differences Between Anatomical Parts of Plants

Anatomical parts that play different roles in a plant are characterized by different structures and, therefore, chemical compositions. For this reason, the PCs in individual plant organs (cones, leaves, and stalks) were also analyzed. In this study, emphasis was placed on the content found in the lower plant level, which is considered the most mature. The total phenolic compound content was highest in the cones of Marynka (357.59 mg/100 g; see Table 1) and Lubelski (271.90 mg/100 g; see Table 1), except for Magnum, where it was higher in the leaves (124.68 mg/100 g; see Table 4). Similar trends were observed in the total flavonoid contents, with Marynka and Lubelski cones exhibiting the highest contents (274.31–211.70 mg/100 g; see Table 1), while in Magnum, the leaves surpassed the cones (89.93 mg/100 g; see Table 4). Specifically, the cones contained two flavonoids: RUT (15.68–184.18 mg/100 g; see Table 1) and K-3-G (7.22–78.46 mg/100 g; see Table 1). The same flavonoids were found in leaves and stalks. In the leaves, K-3-G (50.04–91.21 mg/100 g; see Table 4) was the predominant flavonoid, which was followed by RUT (19.80–62.65 mg/100 g; see Table 4). Conversely, in the stalks, the content of RUT was higher (16.12–26.10 mg/100 g; see Table 7) than that of K-3-G (6.75–8.68 mg/100 g; see Table 7); however, they exhibited lower flavonoid levels compared to the cones and leaves. Among the PAs, GAL exhibited the highest content in the leaves, with 19.45 mg/100 g in Marynka and 8.58 mg/100 g in Magnum (Table 4), while CA was highest in Lubelski (14.30 mg/100 g; see Table 4). In turn, for the cones, the main PAs were caffeic and CGA (in Marynka; see Table 1) as well as CA and SA (in Lubelski and Magnum). In the stalks, the dominant PA was found to depend on the cultivar. The main PA in Marynka was GNT (12.07 mg/100 g; see Table 7), in Lubelski was CA (14.57 mg/100 g; see Table 7), and in Magnum was GAL (16.28 mg/100 g; see Table 7). Instead, the content of GNT in Lubelski exhibited a low level (4.27 mg/100 g; see Table 7). Regardless of these results, the stalks exhibited clearly lower concentrations of PCs than the leaves and cones. According to Jelinek et al. [18], GNT is one of the dominant phenolics in cones from 8-year-old hop plants, with results ranging from 120 mg/100 g to 142.76 mg/100 g for the Saaz cultivar and from 47.25 mg/100 g to 76.89 mg/100 g for the Sladek cultivar, depending on the growth location [18]. Lower contents of PAs were observed in Magnum, ranging from 0.73 to 1.43 mg/100 g for FA and from 1.31 to 1.97 mg/100 g for 4CA (Table 7, Table 8 and Table 9). The presence of PAs in the stalks can be attributed to the high concentration of cell wall polysaccharides, which strengthen this organ. PAs like FA and 4CA are incorporated into the cell wall and play a significant role in cell wall extensibility. FA and its oligomers are vital components of the plant cell walls, forming cross-links between polysaccharides, polysaccharides and lignin, as well as polysaccharides and proteins [23].

### 2.3. Differences Between Different Parts of Plants (Bottom, Middle, and Upper Plant Levels)

The content of PCs was analyzed in three different parts of the plant: bottom (1–3 m above the ground), middle (3–5 m), and upper (5–7 m). These levels reflect the growth of the plant: the highest-level parts are the youngest. In the leaves and cones, it was observed that in the higher-level (younger) parts, there was a significant increase in the content of PCs. In the case of the leaves, this increase ranged from 124.68 to 246.94 mg/100 g in the lower parts to the range 631.17–922.03 mg/100 g in the upper parts of the plants (Table 4, Table 5 and Table 6). For the cones, the PC contents increased from the range 61.68–357.59 mg/100 g (lower hops cones) to 150.39–410.82 mg/100 g in the upper parts (Table 1, Table 2 and Table 3). This comparison shows that the most dynamic changes in the content of PCs took place in the leaves. Looking at individual PCs, very high contents of PAs were recorded in the upper leaves for CGA (45.09 mg/100 g in Marynka) and CA (96.26 mg/100 g in Magnum; see Table 6). In the middle leaves, CGA was present at a high concentration in Lubelski (35.35 mg/100 g; see Table 5), and in the group of flavonoids K-3-G stood out in the upper leaves (244.09–328.62 mg/100 g; see Table 6). In the case of the cones, the differences between plant levels were lower than for the leaves (Table 1, Table 2 and Table 3). Higher contents of RUT and K-3-G were observed in the middle and the upper parts’ leaves and cones in Marynka and Lubelski, whereas in Magnum, they were high only in the middle and upper parts’ leaves. A similar dependence was described by Panwar et al. [27] where a higher content of RUT was observed in in the upper parts in the leaves and fruits of buckwheat.

All hop varieties were characterized by the content of two flavonoids, RUT and K-3-G, which was present especially in the cones and in leaves. The higher contents of RUT and K-3-G were observed in the upper leaves part of Magnum (293.33 mg/100 g and 328.62 mg/100 g, respectively; see Table 6). Instead, the higher contents of these two flavonoids in the cones were in the middle part of Marynka (209.50 mg/100 g and 103.14 mg/100 g, respectively; Table 2). Similar observations regarding RUT were reported by Kobus-Cisowska et al. [19] for the same cultivar analyzed in this study. Additionally, these authors detected the presence of quercetin in the form of aglycone (the free form of RUT).

In the case of the stalks (Table 7, Table 8 and Table 9), the differences between the content of PAs in the lower and upper parts of the plants were moderate, and the highest individual content was recorded for CA (19.54 mg/100 g in middle hops stalks in Lubelski; see Table 8). Differences in the content of PCs in different anatomical parts of the plant and at different degrees of maturity result from the role of these compounds in the plant’s physiology. The content of the PA in question depends on the plant condition; CA and CGA are inhibitors of IAA (indole-3-acetic acid) oxidase, while FA and 4CA are their activators. An increase in IAA may mean that the plant cellular defence system is active against bacteria, viruses, or parasites [28]. Even if 4CA and FA perform opposite functions during plant tissue defence, they are connected through the line of pathway, because the first stands for precursor for FA on the shikimic acid pathway, where the content of the new ones of them is usually higher [29], which was observed especially in leaves independently of hop cultivar (Table 4, Table 5 and Table 6). Analyzing the results, it can be noticed that many factors influence the content of PCs in hops. Paradoxically, the phenolic content in plants may increase in response to unfavorable growing conditions, and both of those are influenced by human activity and conditions occurring naturally. Such conditions can stimulate the enhanced synthesis of these compounds, suggesting that environmental stress may induce an increase in PC levels.

### 2.4. HCA Analysis

The obtained results were additionally analyzed using hierarchical cluster analysis (HCA). This method allowed us to create a matrix of similarities between classified objects (Figure 2). In the current study, HCA confirmed the differences between organs and hop varieties. The upper and middle leaf samples (Magnum, Marynka, and Lubelski) were characterized by large differences from the rest. This group was characterized by a high content of total PCs and a clear dominance of flavonoids over PAs. In turn, on the right side of the plot, there is a stalks cluster, where the content of PCs was the lowest. In the center of the diagram are located mainly samples of cones and lower (most mature) leaves, in which the content of PCs was lower than in young leaves. HCA analysis also showed similarities within the Magnum variety (cones, lower leaves, and upper stalks). In the case of this cultivar, the cones samples were characterized by a significantly lower flavonoid level in comparison to Marynka and Lubelski. The lower leaves of Magnum also contained fewer flavonoids compared to other varieties, and in its upper stalks, PAs clearly dominated over flavonoids.

## 3. Hop Waste Valorization—Phenolic Compounds in Leaves, Stalks, and Small-Caliber Cones

One the goals of our study is to show that the waste generated during hops cultivation can be also a source of important PCs [30]. For this reason, we collected the weight data of the Lubelski cultivar to calculate how many individual PCs can be obtained from one hectare of cultivation.

The collected data presented in Table 10, Table 11 and Table 12 show that the major phenolics sources are leaves, then cones and stalks. The highest content of PAs was recorded in cones (Table 11), where all mentioned compounds were characterized by a higher quantity except for gallic acid and CGA, which were dominant PAs in the leaves (Table 10). With regard to flavonoids, the highest quantity was observed in the leaves, where kaempferol was dominant (955.51 g/ha; see Table 10), but RUT was lower in the cones by almost half (358.67 g/ha and 635.37 g/ha presented in Table 10 and Table 11, respectively).

The poorest sources of PCs were the stalks (822.41 g/ha; see Table 12), even though their weight per hectare as hop waste was the highest. The contents of PAs and flavonoids were lower than in the leaves and cones by about one-half and one-third, respectively (Table 10, Table 11 and Table 12).

Subdivision of each anatomical part of the hop plant into three levels provides a larger amount of data, which can be useful for better waste management. The highest contents of PAs and flavonoids were recorded in the upper and middle leaves and cones (1295.54 g/ha and 715.33 g/ha presented in Table 10 and 1009.90 g/ha and 694.74 g/ha presented in Table 11, respectively); on the other hand, the amount was lower in the bottom parts of the stalks (131.23 g/ha; see Table 12). RUT and K-3-G were the dominant PCs in all anatomical parts from each level (from 18.63 g/ha to 344.23 g/ha for RUT and from 64.31 g/ha to 574. 27 g/ha, respectively; see Table 10 and Table 11) apart from the stalks, where the quantity of K-3-G was lower than GAL and SA from the middle level (Table 12). Instead, the CA content was higher among PAs regardless of the plant level. A similar trend was observed for the PAs, where only SA in the stalks was higher than in the leaves and GAL in the cones (Table 10, Table 11 and Table 12).

The authors in [31] reported that the total phenolic compounds (TPCs) obtained by the FC method in hop leaves was equivalent to 6.22 mg GAE/g (considering our data, we estimate that the TPC yield is 2927.70 g/ha). The mentioned result is much lower than our result (7058.10 g GAE/ha) obtained by the FC method. That differences may be caused by the stage of harvesting (May for the results from Muzykiewicz et al. [31] and September in our case), hop cultivar, and extraction method (98% methanol).

Analysis conducted by [32] using the LC-MS/MS (liquid chromatography–tandem mass spectrometry) method to determinate the phenolic content in the leaves and stalks of Humulus japonicus resulted in yields of 3089.90 mg/kg and 1313.90 mg/kg, respectively (estimated quantity per hectare and considering that our data are equivalent to 1453.92 g/ha and 817.34 g/ha, respectively). Although Humulus japonicus is not commonly used in brewing, the phenolic content in its stalks is comparable to our findings (822.41 g/ha). However, the quantity of K-3-G was reported at 99.9 mg/kg (62.14 g/ha), which is higher than the 53.33 g/ha yield observed in the Lubelski cultivar. Regarding the PAs in Humulus japonicus, the authors claimed 60.70 mg/kg (28.56 g/ha) for the leaves and 283.80 mg/kg (176.54 g/ha) in the stalks for FA, while the quantity of CA came out to 430.10 mg/kg (202.38 g/ha) for the leaves and 256.4 mg/kg (159.50 g/ha) for the stalks. Those results indicate that Humulus japonicus cultivar is a richer source of FA and CA depending on its anatomical part compared to the Lubelski cultivar. On the other hand, in the case of CA in the leaves, it must be considered that the result shown was obtained by quantification of its derivatives, which were identified by a tandem mass spectrometry detector.

In the cones, RUT was the dominant phenolic compound found in Lubelski (635.37 g/ha). The same was analyzed using the UPLC (ultra-high-performance liquid chromatography) method by Bilska et al. [16], where RUT presented as the main phenolic compound equivalent to 967.87 ug/g (540.65 g/ha), which is a similar result to ours. On the contrary, for the PAs, there were significant differences.

The most abundant PA recorded in paper by Bilska et al. [16] was CGA 191.41 ug/g (105.90 g/ha), CA was present in the lowest quantity equivalent to 2.22 ug/g (1.23 g/ha). In turn, while in our study, we reported that the last referred compound was higher as a PA (171.77 g/ha).

The significant differences described regarding the PC quantities may show the impacts of different extraction methods and hops varieties, as well as the region and conditions of the plant growing. However, hop waste in the form of leaves and to a lesser extent stalks is a valid source of PCs. The same should be true for rejected cones due to their small caliber under the minimum considered by the brewing industry [33].

Awareness of the high content of PCs in leaves as well as in other anatomical plant parts brings more useful management to agricultural and urban biomass, which are mainly composted or used as biomass for energy production [34,35], because high contents in lignin in green waste could lead to releases of CO_2_ that are not proportional quantities with respect to the energy obtained. Law changes as a result of responsible management of energy sources due to climate change and depleting fossil fuel resources encourage the search for new sources of compounds that are indispensable in branches of the chemical industry, particularly the building and pharmaceutical industries [36]. Reviewing the literature, there are no limits to the species of plant available and the use of their anatomical parts for products, which can be submitted to their valorization. One of the developing methods used to obtain precious compounds such as bio-oils is pyrolysis. The authors in [37] analyzed three anatomical parts of L. styraciflua (barks, branches, and leaves) from an urban environment in Chile, and their results claimed that this heating process can even increase the quantity of phenol compounds found in bio-oil, especially from bark, which at the same time is the richest source of lignin.

Another example of waste management is the valorization of olive tree leaves and fruit by-products obtained during olive oil production. The majority of waste generated from olive oil production is simply burned without energy recovery or after drying, and this waste is then spread into fields [38]. Olive pomace is a rich source of PCs, yielding up to 100 mg/g [39]. Olive leaves may be also a good source of PCs, and their quantity varies from 42.35 mg/g to 190.65 mg/g depending on the variety [40], but compared to hop leaves, this amount is still low.

Food losses due to overproduction and food not consumed by consumers have a significant impact on the energy losses incurred in their production. Introducing new solutions to recover these losses by adding value to by-products by extracting valuable nutrients from them could provide a new branch of economic value.

## 4. Materials and Methods

### 4.1. Plant Cultivation Conditions

The study was done using three popular Polish cultivars of hops (*Humulus lupulus* L.)—Lubelski, Magnum, and Marynka—which were grown at RZD Jastków, Lublin Province (51°18′12″ N 22°26′12″ E). The plantation was 18 years old at the time of study. The plant material was obtained on 21 September in 2022. The agro-technology of hop cultivation followed the best practice standards. The plants were fertilized twice with NPK: (1) on 27 April 2022 with ‘Gramed’ at a dose of 500 kg/ha, containing 34.52 kg N, 17.26 kg P_2_O_5_, and 40.44 kg K_2_O per 1 hectare; (2) on 8 June 2022 with ‘Pulan Ammonium Sulphate 34.4% N at a dose of 200 kg/ha, containing 59.37 kg N/ha. Additionally, the first application on 27 April 2022 was supplemented with leaf fertilization by boric acid at a dose of 7 kg/ha to deliver boron straight to the stems and leaves of developing and fast-growing plants. On 7–8 July 2022 and 2–3 August 2022, the plants were sprayed with ‘Florovit Agro Liquid Universal’ (Inco SA, Warsaw, Poland) at a dose of 10 l/ha, containing 0.26 kg N and 0.21 kg K_2_O per hectare as well as microelements (Zn, Cu, Mn, and Fe). The sprays were done to strengthen the plants and deliver macro- and microelements directly to the leaves and stems of plants at the stage of cone formation.

### 4.2. Plant Protection

The plant health was monitored throughout the season. The protection of hop plants was composed of five treatments and the three abovementioned applications of liquid fertilizers, which also positively affected the plant health by improving their growing conditions. The direct application of chemical plant protection products was as follows: (1) on 8 June, the application of the fungicide ‘Aliette 80 WG’ at a dose of 2.5 kg/ha; (2) on 23–24 June, insecticide ‘Afinto’ was applied at a dose of 0.18 kg/ha; (3) on 27–28 June, fungicide ‘Aliette 80 WG’ was applied at a dose of 2.5 kg/ha; (4) 7–8 July, fungicide ‘Cuproxat 345 SC’ was applied at a dose of 6 l/ha; (5) on 17–19 August, fungicide ‘Aliette 80 WG’ was applied at a dose of 4 kg/ha. The fungicide ‘Aliette 80 WG’ (Bayer AG, Leverkusen, Germany) contains 80% aluminium fosetyl (800 g/kg). The insecticide ‘Afinto’ produced by Syngenta AG (Basel, Switzerland) contains 500 g/kg of flonicamid to protect hop plants from mites and whiteflies; the compound was used as water suspension. ‘Cuproxat 345 SC’ (Nufarm Ltd., Laverton North, VIC, Australia) contains 190 g/L of tribasic water sulphate in the form of suspension concentrate. No herbicides were used; after pruning the mechanical control of weeds was done four times (21 April, 8–9 June, 30 June–4 July, and 18–20 July), deep tillage was also conducted to make slots in inter-rows through the soil so that the air and water could enter freely into the subsoil and reach deep layers of the root system. Each deep tillage was followed by soil disking or harrowing to level the soil between the rows. All mechanical and chemical treatments were listed in e-agronom, the electronic book for agriculture.

### 4.3. Sampling Principle

Three randomly selected plants were chosen for the study, which normally would be harvested for the beer industry. Each plant was divided into three parts: bottom (1–3 m above the ground), middle (3–5 m), and upper (5–7 m) (Figure 3).

Individual anatomical parts were collected from different levels of the plant, thus obtaining 9 sample organs with different degrees of maturity. The samples were immediately cooled and stored in the freezer until further sample preparation by lyophilization and grinding. The lyophilization and the milling process was carried out successively on freeze dryer Alpha 1-2 LD plus (Osterode am Harz, Germany) and on laboratory grinder IKA A11 Basic (Staufen, Germany) separately for each cultivar, plant level, and plant part. Lyophilization was carried out under pression in −55 °C temperature until the weight did not change. To avoid impurities, between milling procedures, the grinder elements having contact with plant organs were cleaned with MeOH. Each milling procedure was carried out for 30 s to not overheat the sample. The sample (2.5 g) was added to a conical flask (250 cm^3^) which was then filled with 80% 100 cm^3^ MeOH for flavonoids extraction carried out for 2 h on EnviSense laboratory shaker Tos-4030FD (Eindhoven, The Netherlands). Collected extract over the plant sediment was filtered through a disposable and sterile syringe filter (22 µm) directly into HPLC vials and stored in refrigerated conditions for analysis [41]. Room analysis temperature was 20 °C.

### 4.4. Sample Hydrolysis for Phenolic Acid Analysis

The process of samples hydrolysis took place in HCl to obtain 3.42 N (20 cm^3^) solution by laboratory shaking bath Memmert SV1422 WNE14 (Schwabach, Germany) for 205 min at 44.6 °C. After the hydrolysis neutralization of acid by 3.42 N, NaOH (20 cm^3^) was proceeded. Then, a 2-h methanol extraction (100 cm^3^) of PAs was performed at room temperature [42].

### 4.5. Analysis of Phenolic Compounds

The content of PCs was determined by RP-HPLC (Reverse Phase High-Performance Liquid Chromatography) method (Waters, Milford, MA, USA) with gradient, where the mobile phase was a solution of 50% ACN (A)/H_2_O (B) with pH 2.7 (obtained by orthophosphoric acid), which flowed at 1 cm^3^/min though a XBridge Shield column (Waters, Milford, MA, USA) (RP-18, 3.5 µm, b 4.6 × 100 mm) at 30 °C. Each analysis lasted 80 min. It began with a 1-minute isocratic flow (1% A), followed by a linear gradient increase to 50% A from minute 1 to minute 50. From minute 70 to 75, the conditions returned to the initial isocratic flow, while the final 5 min were used for column conditioning. The detection of PCs was done by DAD (Diode Array Detector) (Waters 2998 PDA, Milford, MA, USA) at a spectrum range from 210 nm to 600 nm. All results are expressed in mg/100 g of dry plant mass. PAs and flavonoids were identified on the basis of comparison of their retention times with the retention times of standards and additionally based on UV-Vis spectra (obtained using DAD) [41,42]. The standards of PCs used (Sigma-Aldrich, Saint Louis, MO, USA) were of analytical grade (rutin ≥ 94.0% (HPLC, Sigma-Aldrich, R5143), caffeic acid ≥ 98.0% (HPLC, Sigma-Aldrich, C0625), chlorogenic acid ≥ 95.0% (Sigma-Aldrich C3878), sinapic acid ≥ 98.0% (Sigma-Aldrich, D7927, p-coumaric acid ≥ 98.0% (HPLC, Sigma-Aldrich, C9008), gentisic acid 98.0% (Sigma-Aldrich, 149357), protocatechuic acid ≤ 100.0% (Sigma-Aldrich, P5630), ferulic acid ≥ 98.0% (HPLC, Sigma-Aldrich, 46278), gallic acid 97.5–102.5% (Sigma-Aldrich, G7384) (titration), kaempferol-3-glucoside ≥ 97.0% (HPLC, Sigma-Aldrich, 79851). Calibration curves were prepared from standards corresponding to the identified PCs, and in the case of flavonoid 1 and 2, quercetin was used.

### 4.6. Determination of Total Phenolic Compounds

The total content of PCs was determined by FC method with GAL as the standard. Into the microtitration well plate was applied 2 µL of extracted sample, 58 µL of water, and 10 µL of FC reagent (2 N). The solution was mixed for 5 min at 20 °C, then 30 µL of the saturated Na_2_CO_3_ was added. After 2 h of incubation in laboratory dark, the absorbance was measured at λ = 765 nm by a spectrophotometer (Metertech M960, Taiwan). The TPCs are expressed in milligrams of GAL equivalent per 100 g of dry hop mass.

### 4.7. Number Determination of Lubelski Cultivar Plants per Hectare

The number of plants per hectare was determined by dividing the obtained Lubelski hop mass from a hectare (1650 kg/ha) per medium mass of 10 plants (4.14 kg). The percentual mass relocation in anatomical parts in plants was 33.78%, 28.52%, and 37.70% for cones, leaves, and stalks, respectively. On the base of mentioned data, the yield of PCs per hectare was calculated.

### 4.8. Statistical Analysis

Statistical analysis was performed using one-way analysis variance (ANOVA) with Tukey’s test. All tests were considered significant at *p* < 0.05. Values marked with different letters in the tables are statistically significantly different. To check the similar characteristics between different anatomical part of all hop cultivars in this study, HCA analysis was done. The cluster analysis was carried out on a group of 27 samples (9 per each hop cultivar). Once obtained, we classified data into groups using the agglomeration method, and the one-bond Euclidean distance was measured.

All the calculations were performed by statistical analysis using Statistica (data analysis software system), version 13 (TIBCO Software Inc., Palo Alto, CA, USA, 2017).

## 5. Conclusions

The concept of this work based on qualitative and quantitative analysis. The results presented in Table 1, Table 2, Table 3, Table 4, Table 5, Table 6, Table 7, Table 8, Table 9, Table 10, Table 11 and Table 12 and Figure 1 and Figure 2 show that each anatomical part and position in the plant (level) contain different PC quantities both within the same cultivar and between cultivars, which reflects not only the organ maturity degree but also the exposure to environment factors.

The richest PC sources were the cones and leaves from the Marynka and Lubelski cultivars, while Magnum’s stalks and upper leaves were abundant in phenolics acid.

Lubelski’s examined anatomical parts showed that the richest source of TPC and flavonoids in its leaves (1295.54 g/ha and 1052.04 g/ha, 1009.90 g/ha; see Table 10) followed by its cones (780.85 g/ha, respectively; see Table 11) from the upper level.

The results presented will help to utilize hop by-products (such as leaves and stalks), which constitute a significant biomass content during the production of cones for the brewing industry. Such an approach contributes to aligning production with the zero waste principle and increasing profitability [43,44]. In this context, a promising source of PCs are middle and upper hops leaves, with the dominant compounds being RUT and K-3-G. On the other hand, knowledge of the quantitative and qualitative PC contents is important for composting of hop by-products. PCs can affect gases emission and humic acid synthesis during this process. In order to develop new methods of hop by-products valorization, further research on other secondary metabolites present in this plant are needed.

## Figures and Tables

**Figure 1 molecules-30-02365-f001:**
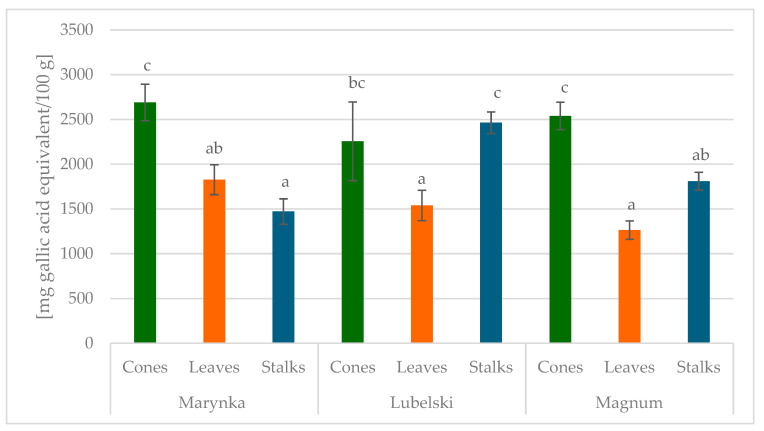
Bar graph of total content of phenolic compounds in analyzed hops anatomical part. Values denoted with different letters differ statistically significantly (*p* ≤ 0.05).

**Figure 2 molecules-30-02365-f002:**
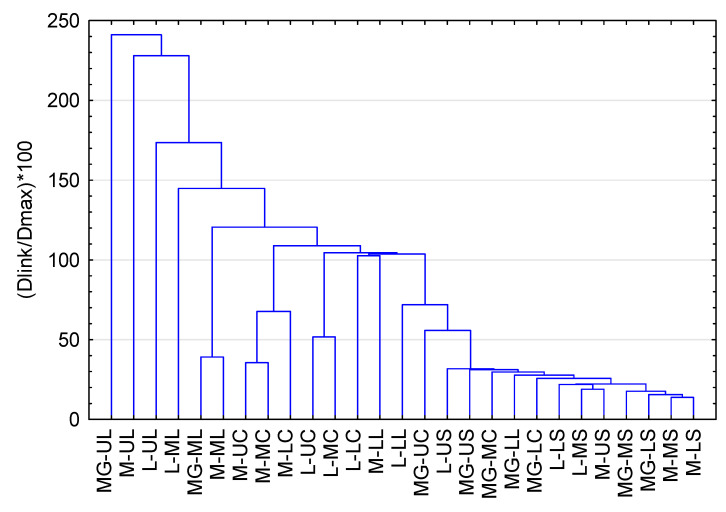
Hierarchical cluster analysis obtained based on the content of phenolic compounds (mg/100 g) in studied samples. Marynka—M; Lubelski—L; Magnum—MG; lower hops cones—LC; middle hops cones—MC; upper hops cones—UC; lower hops leaves—LL; middle hops leaves—ML; upper hops leaves—UL; lower hops stalks—LS; middle hops stalks—MS; upper hops stalks—US.

**Figure 3 molecules-30-02365-f003:**
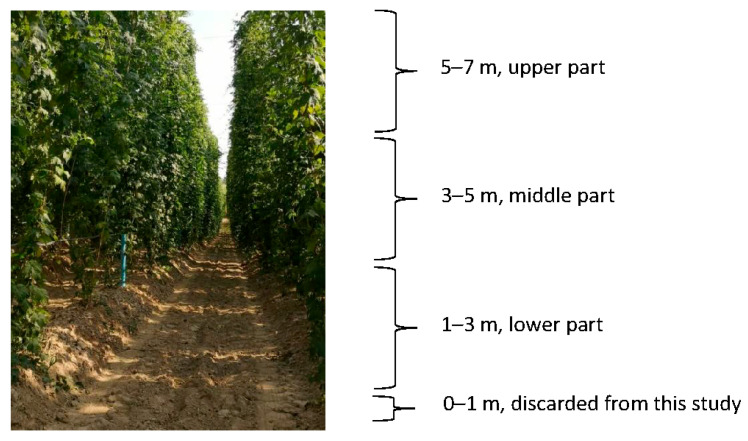
The plants of hop (*Humulus lupulus* L.) at RZD Jastków, Lublin Province, Poland, before harvest.

**Table 1 molecules-30-02365-t001:** The content of phenolic compounds in lower hops cones. Values denoted with different letters differ statistically significantly (*p* ≤ 0.05).

Phenolic Compounds [mg/100 g]	Marynka	Hop Variety Lubelski	Magnum
Gallic acid	7.03 ± 1.60 a	6.10 ± 1.24 a	2.38 ± 0.20 b
p-coumaric acid	6.79 ± 0.55 c	4.77 ± 0.69 b	2.32 ± 0.35 a
Ferulic acid	5.44 ± 0.12 a	7.14 ± 3.26 a	0.84 ± 0.18 a
Caffeic acid	29.03 ± 3.80 c	15.60 ± 5.09 b	4.23 ± 0.48 a
Protocatechuic acid	3.65 ± 0.83 a	5.58 ± 0.64 b	1.23 ± 0.11 a
Chlorogenic acid	23.23 ± 0.76 c	9.24 ± 3.72 b	2.19 ± 0.41 a
Sinapic acid	8.06 ± 0.78 a	14.16 ± 1.28 b	8.63 ± 1.13 a
Rutin	184.18 ± 5.19 c	78.62 ± 2.32 b	15.68 ± 1.39 a
Kaempferol-3-glu	78.46 ± 1.58 a	57.69 ± 0.45 a	7.22 ± 0.45 b
Phenolic acid—total	83.27 ± 9.54 c	60.20 ± 9.36 b	30.68 ± 0.92 a
Flavonoids—total	274.31 ± 7.07 c	211.70 ± 45.66 b	31.00 ± 0.45 a
Phenolic compounds—total	357.59 ± 13.48 c	271.90 ± 54.31 b	61.68 ± 15.71 a

**Table 2 molecules-30-02365-t002:** The content of phenolic compounds in middle hops cones. Values denoted with different letters differ statistically significantly (*p* ≤ 0.05).

Phenolic Compounds [mg/100 g]	Marynka	Hop Variety Lubelski	Magnum
Gallic acid	7.77 ± 1.01 b	10.74 ± 0.47 c	2.07 ± 0.62 a
p-coumaric acid	6.23 ± 0.81 a	7.53 ± 1.09 a	3.25 ± 0.36 b
Ferulic acid	6.45 ± 0.42 a	13.31 ± 4.84 a	7.15 ± 1.40 a
Caffeic acid	31.13 ± 5.19 b	42.91 ± 1.61 c	9.11 ± 2.38 a
Protocatechuic acid	4.04 ± 0.52 b	5.58 ± 0.24 c	1.07 ± 0.32 a
Chlorogenic acid	28.80 ± 0.32 c	18.89 ± 0.21 b	5.19 ± 0.27 a
Sinapic acid	5.14 ± 1.16 b	11.79 ± 1.90 a	10.90 ± 1.05 a
Rutin	209.50 ± 5.75 c	121.63 ± 1.21 b	18.89 ± 0.51 a
Kaempferol-3-glu	103.14 ± 2.94 a	72.36 ± 0.10 a	23.66 ± 1.31 b
Phenolic acid—total	89.57 ± 9.52 a	110.79 ± 8.19 a	47.46 ± 6.67 b
Flavonoids—total	331.78 ± 7.95 a	253.49 ± 5.08 a	55.30 ± 3.39 b
Phenolic compounds—total	421.35 ± 18.07 a	364.29 ± 5.63 a	102.76 ± 9.79 b

**Table 3 molecules-30-02365-t003:** The content of phenolic compounds in upper hops cones. Values denoted with different letters differ statistically significantly (*p* ≤ 0.05).

Phenolic Compounds [mg/100 g]	Marynka	Hop Variety Lubelski	Magnum
Gallic acid	7.55 ± 0.64 c	4.79 ± 0.68 b	1.58 ± 0.42 a
p-coumaric acid	7.36 ± 2.49 a	6.38 ± 0.74 a	5.11 ± 0.40 a
Ferulic acid	5.25 ± 0.17 a	11.08 ± 8.40 a	1.60 ± 0.11 a
Caffeic acid	40.01 ± 8.28 b	29.64 ± 3.62 ab	22.90 ± 0.50 a
Protocatechuic acid	3.92 ± 0.33 c	2.49 ± 0.35 b	0.82 ± 0.12 a
Chlorogenic acid	27.43 ± 4.56 c	18.12 ± 0.57 b	11.38 ± 0.44 a
Sinapic acid	5.71 ± 2.00 a	9.69 ± 0.91 b	8.25 ± 1.47 ab
Rutin	197.62 ± 11.71 c	123.52 ± 0.66 b	43.17 ± 2.18 a
Kaempferol-3-glu	86.30 ± 2.28 b	74.48 ± 0.46 ab	36.92 ± 6.30 a
Phenolic acid—total	97.27 ± 7.36 b	82.22 ± 7.71 ab	63.15 ± 1.72 a
Flavonoids—total	131.55 ± 43.94 a	285.03 ± 4.11 a	87.23 ± 14.31 b
Phenolic compounds—total	410.82 ± 40.10 c	367.25 ± 35.77 b	150.39 ± 13.57 a

**Table 4 molecules-30-02365-t004:** The content of phenolic compounds in lower hops leaves. Values denoted with different letters differ statistically significantly (*p* ≤ 0.05).

Phenolic Compounds [mg/100 g]	Marynka	Hop Variety Lubelski	Magnum
Gallic acid	19.45 ± 0.32 b	7.53 ± 2.39 a	8.58 ± 2.21 a
p-coumaric acid	4.12 ± 0.43 b	0.87 ± 0.22 a	1.15 ± 0.38 a
Ferulic acid	6.47 ± 0.16 a	7.13 ± 0.85 a	3.85 ± 0.05 b
Caffeic acid	19.42 ± 1.42 a	14.30 ± 2.53 a	7.59 ± 3.43 b
Protocatechuic acid	10.11 ± 0.16 b	3.98 ± 1.24 a	4.46 ± 1.15 a
Chlorogenic acid	2.43 ± 0.41 a	10.00 ± 0,14 a	5.68 ± 2.61 a
Sinapic acid	4.34 ± 0.32 a	3.60 ± 0.78 a	3.43 ± 0.72 a
Rutin	62.65 ± 0.51 b	19.80 ± 2.67 a	27.07 ± 0.01 a
Kaempferol-3-glu	91.21 ± 3.46 a	75.66 ± 9.37 a	50.04 ± 0.31 a
Phenolic acid—total	68.51 ± 2.19 a	47.36 ± 1.99 b	34.75 ± 7.55 c
Flavonoids—total	178.43 ± 2.11 a	120.27 ± 17.35 a	89.93 ± 0.13 a
Phenolic compounds—total	246.94 ± 2.94 a	167.63 ± 15.12 ab	124.68 ± 2.31 a

**Table 5 molecules-30-02365-t005:** The content of phenolic compounds in middle hops leaves. Values denoted with different letters differ statistically significantly (*p* ≤ 0.05).

Phenolic Compounds [mg/100 g]	Marynka	Hop Variety Lubelski	Magnum
Gallic acid	12.50 ± 1.95 a	17.85 ± 0.78 b	11.17 ± 1.01 a
p-coumaric acid	1.77 ± 0.46 a	2.56 ± 0.62 a	4.40 ± 0.07 b
Ferulic acid	8.18 ± 0.49 a	12.11 ± 1.03 b	15.56 ± 1.42 c
Caffeic acid	16.22 ± 0.38 b	23.82 ± 3.04 a	24.73 ± 1.26 a
Protocatechuic acid	6.50 ± 1.01 a	9.28 ± 0.40 b	5.81 ± 0.52 a
Chlorogenic acid	15.64 ± 6.53 a	35.35 ± 2.11 b	19.89 ± 2.34 a
Sinapic acid	3.98 ± 0.38 a	5.12 ± 1.56 a	4.59 ± 1.72 a
Rutin	139.09 ± 8.20 a	74.93 ± 7.05 b	133.18 ± 14.89 a
Kaempferol-3-glu	126.90 ± 7.61 a	219.63 ± 15.59 a	191.33 ± 15.58 a
Phenolic acid—total	64.80 ± 7.41 a	106.10 ± 2.69 b	86.15 ± 3.72 c
Flavonoids—total	314.02 ± 9.42 a	412.66 ± 43.47 a	357.17 ± 9.37 a
Phenolic compounds—total	378.82 ± 13.55 a	518.77 ± 42.95 a	443.33 ± 11.87 a

**Table 6 molecules-30-02365-t006:** The content of phenolic compounds in upper hops leaves. Values denoted with different letters differ statistically significantly (*p* ≤ 0.05).

Phenolic Compounds [mg/100 g]	Marynka	Hop Variety Lubelski	Magnum
Gallic acid	15.99 ± 0.98 a	14.23 ± 0.71 a	18.41 ± 4.01 a
p-coumaric acid	3.80 ± 0.02 a	4.11 ± 0.77 a	16.69 ± 3.37 c
Ferulic acid	12.59 ± 0.21 a	8.40 ± 1.47 a	12.45 ± 2.86 a
Caffeic acid	39.08 ± 3.55 a	30.91 ± 3.45 a	96.26 ± 17.02 b
Protocatechuic acid	8.31 ± 0.51 a	7.40 ± 0.36 a	9.58 ± 2.08 a
Chlorogenic acid	45.09 ± 1.89 b	34.02 ± 1.38 a	59.15 ± 1.30 c
Sinapic acid	3.31 ± 0.30 a	4.43 ± 1.57 a	3.34 ± 1.22 a
Rutin	241.78 ± 12.24 b	99.57 ± 13.43 a	293.33 ± 14.61 c
Kaempferol-3-glu	296.78 ± 18.48 a	244.09 ± 21.48 b	328.62 ± 5.35 a
Phenolic acid—total	165.25 ± 10.08 b	107.44 ± 6.44 a	215.90 ± 27.07 c
Flavonoids—total	586.32 ± 17.63 b	523.72 ± 37.24 a	706.13 ± 15.34 c
Phenolic compounds—total	751.57 ± 23.08 b	631.17 ± 44.46 a	922.03 ± 43.29 c

**Table 7 molecules-30-02365-t007:** The content of phenolic compounds in lower hops stalks. Values denoted with different letters differ statistically significantly (*p* ≤ 0.05).

Phenolic Compounds [mg/100 g]	Marynka	Hop Variety Lubelski	Magnum
Gallic acid	2.48 ± 0.21 a	4.61 ± 0.20 a	16.28 ± 2.44 b
p-coumaric acid	1.48 ± 0.47 a	1.73 ± 0.15 a	1.31 ± 0.11 a
Ferulic acid	3.18 ± 1.19 b	1.12 ± 0.20 a	1.39 ± 0.30 ab
Caffeic acid	9.16 ± 1.22 a	14.57 ± 2.19 b	7.50 ± 0.32 a
Protocatechuic acid	2.51 ± 0.22 a	2.74 ± 0.67 a	4.00 ± 0.32 b
Gentisic acid	12.07 ± 1.16 a	4.27 ± 0.36 b	10.48 ± 0.50 a
Chlorogenic acid	6.29 ± 1.15 a	6.83 ± 0.61 a	6.33 ± 1.42 a
Rutin	16.12 ± 1.33 a	26.10 ± 1.66 b	17.77 ± 0.67 a
Kaempferol-3-glu	8.68 ± 0.11 a	7.64 ± 0.79 a	6.75 ± 1.21 a
Phenolic acid—total	37.17 ± 3.91 a	35.87 ± 3.06 a	47.29 ± 1.64 b
Flavonoids—total	24.81 ± 1.37 a	40.11 ± 2.67 b	22.18 ± 2.01 a
Phenolic compounds—total	61.98 ± 5.24 a	78.98 ± 4.47 b	68.47 ± 3.63 a

**Table 8 molecules-30-02365-t008:** The content of phenolic compounds in middle hops stalks. Values denoted with different letters differ statistically significantly (*p* ≤ 0.05).

Phenolic Compounds [mg/100 g]	Marynka	Hop Variety Lubelski	Magnum
Gallic acid	2.89 ± 0.33 a	9.23 ± 0.80 b	16.23 ± 1.32 c
p-coumaric acid	1.91 ± 0.28 a	2.49 ± 0.09 b	1.97 ± 0.26 ab
Ferulic acid	3.54 ± 0.18 c	1.52 ± 0.03 b	0.73 ± 0.12 a
Caffeic acid	16.05 ± 3.24 a	19.54 ± 0.95 b	12.65 ± 0.49 a
Protocatechuic acid	3.26 ± 0.19 b	4.21 ± 0.24 a	4.64 ± 0.27 a
Gentisic acid	10.31 ± 0.07 c	3.60 ± 0.25 a	6.12 ± 0.34 b
Chlorogenic acid	11.63 ± 2.04 a	9.02 ± 1.30 a	10.52 ± 2.67 a
Rutin	15.11 ± 1.83 a	28.31 ± 2.51 c	22.81 ± 1.84 b
Kaempferol-3-glu	7.75 ± 0.22 a	8.35 ± 0.92 a	3.72 ± 1.14 b
Phenolic acid—total	49.59 ± 3.22 a	40.58 ± 1.73 a	52.91 ± 2.70 a
Flavonoids—total	22.86 ± 2.03 a	41.28 ± 6.28 b	27.38 ± 3.14 a
Phenolic compounds—total	72.45 ± 4.60 a	81.86 ± 7.72 b	80.29 ± 2.18 ab

**Table 9 molecules-30-02365-t009:** The content of phenolic compounds in upper hops stalks. Values denoted with different letters differ statistically significantly (*p* ≤ 0.05).

Phenolic Compound [mg/100 g]	Marynka	Hop Variety Lubelski	Magnum
Gallic acid	3.93 ± 0.75 a	5.64 ± 1.30 a	25.97 ± 7.29 b
p-coumaric acid	2.29 ± 0.18 a	1.87 ± 0.07 a	4.24 ± 0.46 b
Ferulic acid	5.26 ± 0.16 c	2.43 ± 0.08 b	1.43 ± 0.11 a
Caffeic acid	14.74 ± 4.30 a	17.84 ± 1.22 a	13.53 ± 0.51 a
Protocatechuic acid	3.52 ± 0.34 a	2.97 ± 0.18 a	3.61 ± 0.94 a
Gentisic acid	14.49 ± 0.23 b	7.38 ± 3.35 a	5.92 ± 0.10 a
Chlorogenic acid	9.01 ± 3.42 ab	7.21 ± 0.66 a	14.14 ± 2.72 c
Rutin	29.23 ± 8.05 ab	40.93 ± 0.42 b	27.13 ± 1.84 a
Kaempferol-3-glu	16.15 ± 4.84 a	9.08 ± 3.28 a	9.01 ± 1.42 a
Phenolic acid—total	53.24 ± 0.67 a	44.98 ± 0.88 a	68.84 ± 10.44 b
Flavonoids—total	45.38 ± 12.89 ab	60.62 ± 8.73 b	36.14 ± 4.26 a
Phenolic compounds—total	98.62 ± 12.30 a	105.60 ± 9.48 a	104.98 ± 14.31 a

**Table 10 molecules-30-02365-t010:** The yield of phenolic compounds in Lubelski variety per 1 ha in upper, middle, and lower leaves.

Phenolic Compounds [g/ha]	Gallic Acid	p-Coumaric Acid	Ferulic Acid	Caffeic Acid	Protocatechuic Acid	Chlorogenic Acid	Sinapic Acid	Rutin	Kaempferol-3-glu	Phenolic Acid—Total	Flavonoids—Total	Phenolic Compounds—Total
**Upper**	33.48	9.67	19.76	72.72	17.41	80.04	10.42	234.26	574.27	243.51	1052.04	1295.54
**Middle**	25.20	3.61	17.09	33.62	13.10	49.90	7.23	105.77	310.04	149.76	565.57	715.33
**Lower**	7.09	0.82	6.71	13.46	3.75	9.41	3.39	18.63	71.20	44.62	134.45	179.07
**Total**	65.76	14.10	43.57	119.80	34.26	139.35	21.04	358.67	955.51	437.88	1752.06	2189.94

**Table 11 molecules-30-02365-t011:** The yield of phenolic compounds in Lubelski variety per 1 ha in upper, middle, and lower cones.

Phenolic Compounds [g/ha]	Gallic Acid	p-Coumaric Acid	Ferulic Acid	Caffeic Acid	Protocatechuic Acid	Chlorogenic Acid	Sinapic Acid	Rutin	Kaempferol-3-glu	Phenolic Acid—Total	Flavonoids—Total	Phenolic Compounds—Total
**Upper**	13.35	17.78	30.88	82.60	6.94	50.50	27.00	344.23	207.56	229.05	780.85	1009.90
**Middle**	17.96	12.59	22.26	71.75	9.33	31.59	19.71	203.38	120.99	185.19	509.56	694.74
**Lower**	6.80	5.32	7.96	17.39	6.22	10.30	15.78	87.64	64.31	69.77	221.72	291.49
**Total**	38.11	35.70	61.10	171.77	22.49	92.40	62.52	635.37	392.94	484.10	1512.41	1996.51

**Table 12 molecules-30-02365-t012:** The yield of phenolic compounds in Lubelski variety per 1 ha in upper, middle, and lower stalks.

Phenolic Compounds [g/ha]	Gallic Acid	p-Coumaric Acid	Ferulic Acid	Caffeic Acid	Protocatechuic Acid	Chlorogenic Acid	Sinapic Acid	Rutin	Kaempferol-3-glu	Phenolic Acid—Total	Flavonoids—Total	Phenolic Compounds—Total
**Upper**	17.54	5.82	7.56	55.49	9.24	22.95	22.43	127.31	28.24	141.02	296.57	437.60
**Middle**	17.23	4.65	2.84	36.47	7.86	6.72	16.83	52.83	15.58	92.58	161.00	253.58
**Lower**	5.74	2.15	1.39	18.13	3.41	5.31	8.50	32.47	9.51	44.63	86.61	131.23
**Total**	40.50	12.62	11.79	110.08	20.50	34.99	47.76	212.61	53.33	278.23	544.18	822.41

## Data Availability

The raw data are available upon reasonable request from the author for correspondence.

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
