# Peer review of "Hop (Humulus lupulus L.) Phenolic Compounds Profile Depends on Cultivar and Plant Organ Maturity"

_molecules, 2025, doi:10.3390/molecules30112365_

Round 1

Reviewer 1 Report

Comments and Suggestions for Authors

The manuscript entitled “Profiles of phenolic compounds in hop (Humulus lupulus L.)  depend on the cultivar and anatomical parts of the plant” is written well but I have some major queries.

  1. Table 7 shows the results of the content of phenolic compounds in lower hops stalks, whereas Supplementary Fig. S12. Chromatogram of lower hop stalks preceded by hydrolysis in Marynka cultivar 690 at 320 nm shows few compounds present in it. Justify?

kindly author labelled the compounds presented in the supplementary figure 12. Also, this is the chromatogram of Marynka cultivar.

Also, include the labelled chromatogram of Lubelski and Magnum variety.

  1. Table 3 shows the results of the content of phenolic compounds in upper hops cones., whereas Supplementary Fig. S13. Chromatogram of upper hop cones in Marynka cultivar at 355 nm shows one or two compounds with very little area response present in it? Justify?

On the other hand, Table 3 shows all compounds present with very good amounts. Justify?

kindly author labelled the compounds presented in the supplementary figure 13. Also, this is the chromatogram of Marynka cultivar.

Also, include the labelled chromatogram of Lubelski and Magnum variety.

  1. Table 5 shows the results of the content of phenolic compounds in middle hops leaves, whereas Supplementary Fig. S14. Chromatogram of middle hop leaves in Lubelski cultivar at 355 nm shows one or two compounds with very little area response present in it. On the other hand, Table 5 shows all compounds present with very good amounts with all variety. Justify?

kindly author labeled the compounds presented in the supplementary figure 14. Also, this is the chromatogram of Lubelski cultivar.

Also, include the labelled chromatogram of Marynka Lubelski and Magnum variety.

  1. In supplementary figure S15 to S23 all the calibration curve shows the wrong regression equation. Also wondering about the R2 shows 1 for all the regression equations. The author should include system generated calibration curves in the main text or as supplementary figures.
  2. The author should include all the chromatograms of hops cones and leaves with upper, middle and lower labelled chromatograms in the main text or as supplementary figures.
  3. The author should include the references for HPLC analysis of phenolic compounds.

Reviewer 2 Report

Comments and Suggestions for Authors

Abstract

- It would be better to add Background section to the Abstract.

- Add sampling principle to Materials and methods section.

- The absolute values presented without standard error of the mean and with an accuracy to hundredths (which raises doubts).

- Rewrite the conclusion (the content… of content – sounds bad).

- It is better to add practical meaning of obtained results instead of declaring absolute values.

Introduction

- Lines 36-36 – add more references. References 1 and 2 are not enough.

- Lines 52-53 – authors describe beneficial effects of… of what? Did you mean polyphenols? You present near six effects, but prove them with only two references, one of which doesn`t have any English translation, at least the anstract.

- I offer to move the aim of the study to the end of the Introduction section.

Materials and methods.

- Subsections 4.1 and 4.2 are overloaded with information which doesn`t correspond to the aim of the work.

- Line 574 – the plants were randomly chosen. Is there any information about choice criteria? Or you mean some specific plant for analysis?

 - Please explain the sampling method. Do you mean that there are nine samples from one plant?

Conclusions

Give a practical solution of your study in the Conclusions section.

Figures and Tables

- Table 1-9. Table data are duplicated in the text in absolute values.

- In general, authors duplicate data presented in tables in text, presenting absolute values.

- Figures also duplicate tables and text. Please make sure, that figures and tables don`t duplicate each other and the data presented in the text.

- Add error bars to the figures.

Reviewer 3 Report

Comments and Suggestions for Authors

The manuscript entitled: Profiles of phenolic compounds in hop (Humulus lupulus L.) depend on the cultivar and anatomical parts of the plant.

Aimed to: To investigate how the profiles and content of phenolic compounds in hop plants (Humulus lupulus L.) vary according to cultivar, anatomical part (cones, leaves, stalks), and vertical level within the plant.

Answering the main question: To what extent do the profiles and levels of phenolic compounds differ across various cultivars of hop, different plant organs, and vertical positions within the plant structure?

The topic is relevant and addresses a specific gap in current knowledge since until now, most research has focused only on cones, and often as average cone biomass without considering plant level (top, middle, bottom). The leaves and stalks, which constitute a large part of the biomass and are typically discarded as by-products, have not been studied in this context. By assessing vertical variation within the plant (due to maturity or light exposure), the study explores an under-investigated dimension that can impact both biological understanding and industrial utilisation.

The novelty lies in to be the first-time analysis of phenolic compound profiles in leaves and stalks of hop plants, not just cones. Presenting a new perspective on compound distribution within a tall plant by analysis of vertical stratification (top, middle, bottom) of phenolic content. Also having implications for zero-waste utilisation of hop by-products (leaves and stalks), which could be valuable sources of bioactive compounds like rutin and kaempferol-3-glucoside. Presenting potential value for sustainable practices in agriculture and brewing, including composting and functional food or cosmetic applications.

Overall, this research is both timely and innovative, bridging a critical knowledge gap in the phytochemical study of hops. It offers practical applications in the context of sustainable agriculture ad bioeconomy. The methodology is well-structured and provides a solid foundation for future studies, particularly on secondary metabolites in non-cone hop tissues. The work has clear potential to influence future practices in hop cultivation, processing, and waste utilisation.

The manuscript is structured, clear and easy to read, but I would like to suggest some changes and improvements.

Please use the Journal template (Molecules).

The title can be restructured to make it easier.

The abstract as more than 200 words but according journal rules must be less.

According to the journal's instructions the abstract presents overall main studies and used methods with the principal results and conclusions.

Keywords (are presenting 5 of 10 possible) The words part of the title doesn't need to be as keyword.

The introduction is well-prepared and relevant to the work.

The methodology is well described, but can be improved with formatting corrections that can be solved and a better description of the used equipment:

For each mentioned equipment please add (brand, city, state abbreviation if USA, country) and its operating conditions (equipments to review: shaking bath, lyophilizer, freezer, grinder, etc)

The first time you use an abbreviation please write it out in full firstly.

Please consider to add error bars to Graphic figures

Tables can have the 1st line with Marynka and Magnum catered in the line.

Please format Table 10 to 12 using Journal rules.

The reader should not need to look in the main text for explanations of abbreviations in tables and figures, so they should have self-explanatory captions and footnotes.

In section 4.3 consider to use different paragraphs, do the separation of the process hydrolysis from the previous treatment.

What do you mean with Waters Asc.? I know Waters Corporation.

Plase add concentration to the solutions used as for example to Na2CO3  section 4.5.

The results and discussion are extensive and well explained, usually, the results obtained are discussed and compared with those from other works

In the units: Please use µ and not u.

The conclusions need to be improved showing they are consistent with the evidence and arguments presented and answer the main question posed.

The reference list needs to be reviewed in terms of the formatting. Species names need to be in italic. The journal title must be in italic but the paper title not. Page numbers and Doi.

References are adequate and in 38 papers presented, only 1 is from some of this manuscript's authors, so manuscript had not auto-citation.

In the supplementary material considers to identify the peaks with a label in the chromatograms and to add the integration wave length.

Please identify the peaks in the chromatograms of Figure S12 to S14.

How can you get R2=1.000 for all the calibrations carried out? Can you add one more decimal? Why do you force the (0,0) point?

Considering the comments an article revision is recommended.

Comments on the Quality of English Language

English is not my mother tongue but I advise an English revision to all manuscript.

The English could be improved to express the research work more clearly and errors in words and/or sentences must be found and solved.

Round 2

Reviewer 1 Report

Comments and Suggestions for Authors

Author significantly responded all the queries.

Reviewer 2 Report

Comments and Suggestions for Authors

In general, authors tried to improve their article. They corrected English language, added some references and substantiated the relevance of their study.

However the Reviewer has some questions left.

  1. Conclusion section disappeared from the Abstract.
  2. The aim of the study is presented in lines 126-127, not in lines 126-139. The same refers to the Abstract.
  3. It is better to place tables and figures immediately after they are mentioned in the text, not at the end of the section.
  4. Authors added many new abbreviations. However the abbreviations list is absent. Some of the abbreviations seem excessive.
